# Bactericidal effect of tetracycline in E. coli strain ED1a may be associated with ribosome dysfunction

Iskander Khusainov [1,7], Natalie Romanov[1], Camille Goemans [2,8], Beata Turoňová [1], Christian E. Zimmerli [1,9], Sonja Welsch [3], Julian D. Langer [4,5], Athanasios Typas [2] & Martin Beck [1,6] ✉

Ribosomes translate the genetic code into proteins. Recent technical advances have facilitated in situ structural analyses of ribosome functional states inside eukaryotic cells and the minimal bacterium Mycoplasma. However, such analyses of Gram-negative bacteria are lacking, despite their ribosomes being major antimicrobial drug targets. Here we compare two *E. coli* strains, a lab *E. coli* K-12 and human gut isolate *E. coli* ED1a, for which tetracycline exhibits bacteriostatic and bactericidal action, respectively. Using our approach for close-to-native E. coli sample preparation, we assess the two strains by cryo-ET and visualize their ribosomes at high resolution in situ. Upon tetracycline treatment, these exhibit virtually identical drug binding sites, yet the conformation distribution of ribosomal complexes differs. While K-12 retains ribosomes in a translation-competent state, tRNAs are lost in the vast majority of ED1a ribosomes. These structural findings together with the proteome-wide abundance and thermal stability assessments indicate that antibiotic responses are complex in cells and can differ between different strains of a single species, thus arguing that all relevant bacterial strains should be analyzed in situ when addressing antibiotic mode of action.

Ribosomes are essential molecular machines responsible for protein synthesis by decoding genetic information from messenger RNA (mRNA) into polypeptide chains. This process requires the assistance of several translation factors and consists of four main stages: initiation, elongation, termination, and recycling. The actual synthesis of new polypeptide chains happens during repetitive rounds of elongation. During this process in bacteria, translation factor EF-Tu delivers aminoacyl-tRNA to the ribosomal A-site, ensuring its proper accommodation on the ribosome, while translation factor EF-G facilitates the translocation of the tRNAs from the A- and P-sites to P- and E-sites respectively, and promotes progression of the ribosome along the mRNA[1]. The P-site tRNA is a critical component of the translation machinery, as it carries the peptide chain and interacts with the ribosome to ensure its correct positioning for the next round of elongation. Recent advances of the in situ cryo-electron tomography (cryo-ET) expanded structural analysis of the ribosomes from in vitro

[1]Department of Molecular Sociology, Max Planck Institute of Biophysics, Max-von-Laue-Straße 3, 60438 Frankfurt am Main, Germany. [2]European Molecular Biology Laboratory, Genome Biology Unit, Meyerhofstraße 1, 69117 Heidelberg, Germany. [3]Central Electron Microscopy Facility, Max Planck Institute of Biophysics, Max-von-Laue-Straße 3, 60438 Frankfurt am Main, Germany. [4]Membrane Proteomics and Mass Spectrometry, Max Planck Institute of Biophysics, Max-von-Laue-Straße 3, 60438 Frankfurt am Main, Germany. [5]Mass Spectrometry, Max Planck Institute for Brain Research, Max-von-Laue-Straße 4, 60438 Frankfurt am Main, Germany. [6]Institute of Biochemistry, Goethe University Frankfurt, 60438 Frankfurt am Main, Germany. [7]Present address: European Molecular Biology Laboratory, EMBL Grenoble, 71 Av. des Martyrs, 38000 Grenoble, France. [8]Present address: School of Life Sciences, École Polytechnique Fédérale de Lausanne (EPFL), SV, Station 19, 1015 Lausanne, Switzerland. [9]Present address: Institute of Physics, École Polytechnique Fédérale de Lausanne (EPFL), BSP Route de la Sorge, 1015 Lausanne, Switzerland. ✉e-mail: martin.beck@biophys.mpg.de

reconstituted complexes with tRNAs and translation factors to visualization of their action in close to native environment and under stress in cells[2–6]. Surprisingly, bacteria remain underrepresented among organisms for which translation states have been characterized at high resolution by cryo-ET and subtomogram averaging (STA). Until now, such studies have been mainly conducted on the minimal bacterium *Mycoplasma pneumonia*, and uncovered that translation inhibitors shift the distribution of ribosome states inside the cells[4,7]. Nonetheless, antibiotic mode-of-action, secondary targets, and resistance can differ substantially across bacterial species, and sometimes strains.

Strikingly, bacteriostatic ribosome-binding antibiotics such as tetracyclines are bactericidal to some members of the gut microbiota, and may thus cause acute damage to this complex community of microorganisms[8]. This means that antibiotics known to only stop bacterial growth of the few model strains tested before, also effectively kill certain gut bacteria[8]. One remarkable example is the gut isolate *E. coli* ED1a[9], which was found to be killed by tetracyclines, in particular doxycycline[8], unlike *E. coli* BW25113 (*E. coli* K-12), one of the best-characterized organisms in molecular biology[10]. Although doxycycline has the same minimal inhibitory concentration (MIC) of 4 μg/ml for both strains[8], they apparently respond to the treatment differently.

It remains unclear why a gut commensal *E. coli* strain would show such a distinct reaction to a drug compared to the lab strain, especially when treated with well-described and established antibiotics such as tetracyclines. The primary target of tetracycline (hereafter referred to as TET) is the A-tRNA binding site of the small subunit of the bacterial ribosome[11], and its bacteriostatic action is mainly attributed to its reversible binding. Structures of the 70S ribosome in complex with TET solved for gram-negative bacteria *E. coli* SQ171 and *T.*

*thermophilus*[12–14], together with their structure alignment to a vacant ribosome of gram-positive *Staphylococcus aureus*[15], suggest that the TET binding mode is overall conserved.

Despite the known primary ribosomal binding site of TET, it is still unknown to which extent TET impacts the translation apparatus inside cells. Specifically, it is unclear if TET shifts translation states, whether ribosomes remain as intact 70S particles, and/or if a pool of ribosomes is potentially protected in a hibernation-like manner[16,17]. Understanding the ribosome dynamics would thus allow further characterization of TET's mode of action in situ. Thus far, in situ structural analysis of translational states has been done in minimal bacteria *Mycoplasma pneumonia*[4], amoeba *Dictyostelium discoideum*[3], and human cells[5], but to the best of our knowledge, is missing in gram negative bacteria.

Here, we apply a number of complementary methods, including thermal stability quantitative proteomics, in situ cryo-electron tomography (cryo-ET), subtomogram averaging (STA), and single particle cryo-electron microscopy (Fig. 1) to investigate how the translation system copes with exposure to TET in the lab strain *E. coli* K-12 in comparison to the gut isolate *E. coli* ED1a. In doing so, we report structures of ribosomes inside both bacterial strains.

## Results

### ED1a accumulates more intracellular TET than K-12
We aimed to further define the differences of the antibiotic stress response between two *E. coli* strains, the laboratory strain *E. coli* BW25113 (K-12), and the gut isolate *E. coli* ED1a at the molecular level. The experimental setup is described in a schematic illustration in Fig. 1. A previous study reported that these two strains exhibit differential

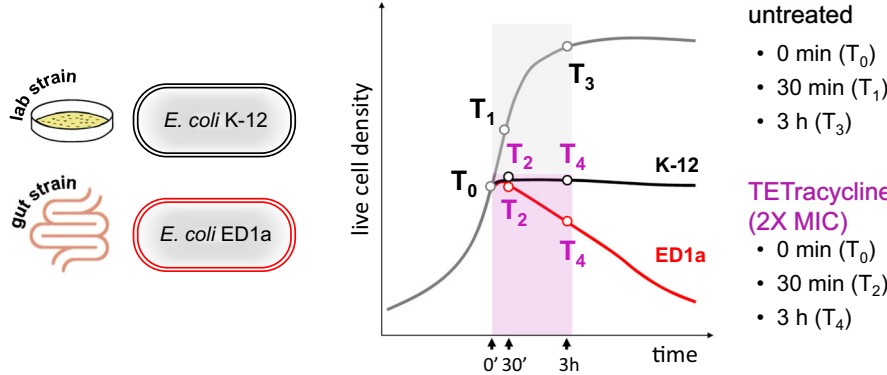

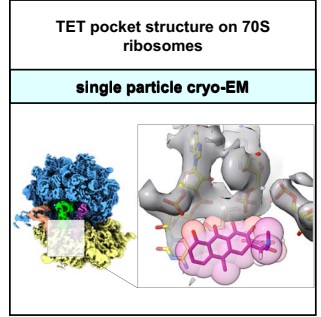
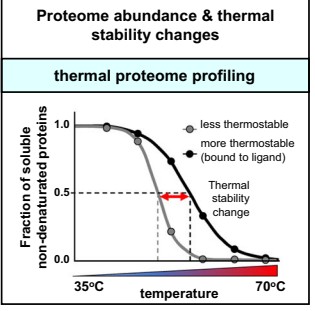
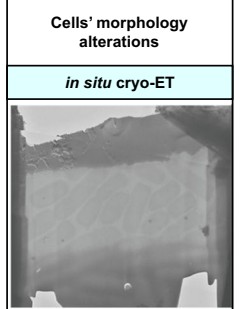
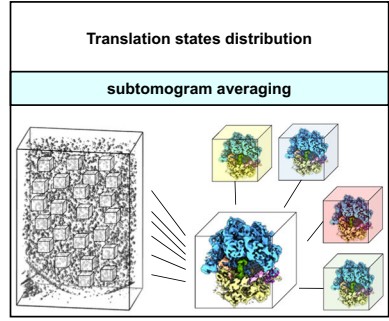

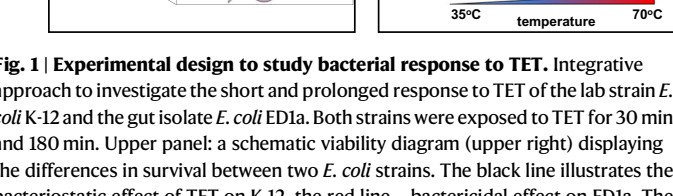

**Fig. 1 | Experimental design to study bacterial response to TET.** Integrative approach to investigate the short and prolonged response to TET of the lab strain *E. coli* K-12 and the gut isolate *E. coli* ED1a. Both strains were exposed to TET for 30 min and 180 min. Upper panel: a schematic viability diagram (upper right) displaying the differences in survival between two *E. coli* strains. The black line illustrates the bacteriostatic effect of TET on K-12, the red line – bactericidal effect on ED1a. The gray line illustrates the growth of untreated cells. White dots represent the time-points used in the study. Lower panel: the structure of the respective TET binding site on the 70S ribosomes was addressed by single particle cryo-EM; changes in proteome abundance and thermal stability by thermal proteome profiling, cell morphology by in situ cryo-ET, and the translation states of ribosomes by sub-tomogram averaging.

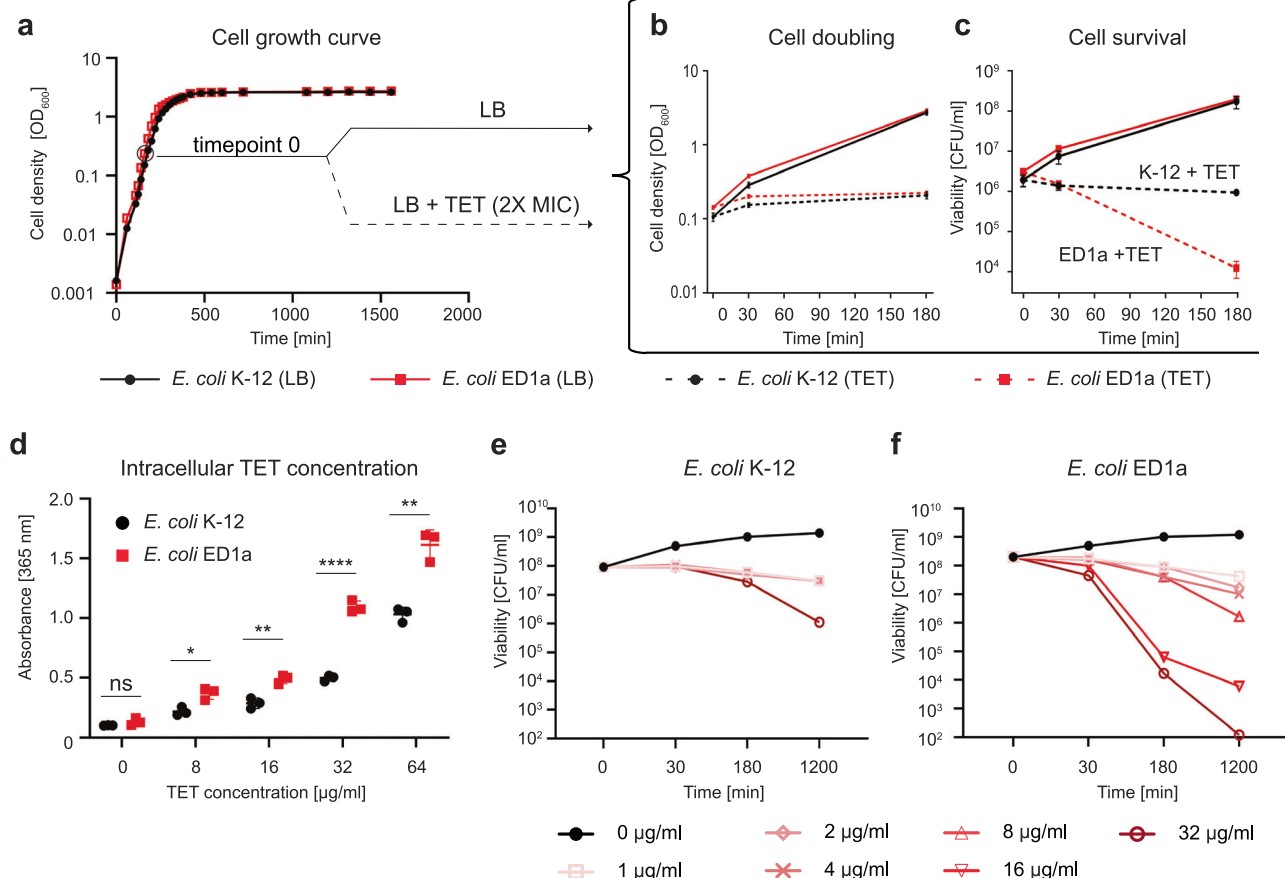

**Fig. 2 | The bactericidal effect is not linked to changes of TET intracellular concentration. a** Growth curves of two *E. coli* strains K-12 (black) and *E. coli* ED1a (red). The profiles show almost identical growth of the two strains. Timepoint 0 indicates the cells at exponential growth phase chosen for treatment with antibiotic in subsequent experiments. Source data are provided as a Source Data file. **b** The ability of cells to double in presence of TET assessed through the optical density measurements of cultures in exponential growth phase in LB (continuous lines) or in LB containing 8 μg/ml, (2X MIC) TET (dashed lines). The cells of both strains were unable to double in presence of TET. Source data are provided as a Source Data file. **c** Viability of cells treated with 8 μg/ml, (2X MIC) TET as in (**b**) assessed by counting colony-forming units (CFU) on LB (plates). The values and error bars represent averages ± two standard deviations for three independent biological replicates for each condition (of three independent measurements at every timepoint of every sample). Source data are provided as a Source Data file. **d** Intracellular TET

concentration of cultures exposed to different TET concentrations (0−64 μg/ml) for 30 min, measured as absorbance of cell lysates at 365 nm. The concentration in *E. coli* K-12 (black) is almost two times lower than the *E. coli* ED1a (red). Statistical significance was calculated using a two-sided unpaired *t* test of measurements obtained from 3 independent biological replicates. The values and error bars represent means ± standard deviations of three biological replicates. The significance symbols indicate *P* values: ns for *P* = 0.2356, * for *P* = 0.0124, ** for *P* = 0.0038, **** for *P* < 0.0001. Source data are provided as a Source Data file. Viability of the *E. coli* K-12 (**e**) and ED1a (**f**) cells exposed to different TET concentrations (0−32 μg/ml) for up to 20 h, assessed by counting colony forming units (CFU) on LB (plates/agar). The survival of *E. coli* ED1a cells is lower than that of *E. coli* K-12. The values and error bars represent averages ± two standard deviations of three biological replicates. Source data are provided as a Source Data file.

responses to the tetracycline derivative doxycycline[8]. For both strains, we observed a similar effect of conventional TET at double minimal inhibitory concentration (2X MIC, which equals to 8 μg/ml for both strains). The number of culturable cells of *E. coli* K-12 did not change subsequent to exposure to TET, suggesting its bacteriostatic effect on the lab strain, while the viability of ED1a cells dropped by almost 99% after 3 h, demonstrating a bactericidal effect of the antibiotic on the gut isolate (Fig. 2a−c). For both strains, we measured the intracellular concentration of TET by spectrophotometry, taking advantage of a high absorbance peak of TET at 365 nm, which is otherwise absent in untreated *E. coli* lysates (Supplementary Fig. 1a). At conditions ranging from 0 to 64 μg/ml of TET added to the medium, the intracellular concentration of antibiotic was about twice higher in ED1a cells compared to K-12 (Fig. 2d). Nevertheless, even at treatment concentrations up to 4X and 8X MIC, the K-12 strain displayed a significantly higher survival rate than the ED1a (Fig. 2e, f). Furthermore, even comparable intracellular TET concentrations (e.g., ED1a treated with 8 μg/ml versus K-12 treated with 16 μg/ml) led to significantly more death in ED1a.

These results suggest that this antibiotic's intracellular concentration difference is not the sole cause of observed viability effects, however, more subtle effects cannot be excluded.

## The known TET-binding pockets are indistinguishable in K-12 and ED1a ribosomes

To test if TET could similarly bind to its target in both strains, we solved three high-resolution single-particle cryo-EM structures. The structure of *E. coli* ED1a 70S vacant ribosome bound to TET, (Fig. 3a, Supplementary Fig. 1b, c) did not exhibit any considerable differences in the TET-binding region compared to the available structures of *E. coli* K-12 70S·TET complexes (Supplementary Fig. 1d, e). Additionally, we did not observe TET density at any of secondary binding sites reported earlier[18].

Since TET targets active ribosomes charged with functional ligands in exponentially growing cells, we also resolved single particle cryo-EM structures of 70S ribosomes bound to mRNA and three tRNAs. In order to obtain endogenously formed complexes, we bypassed

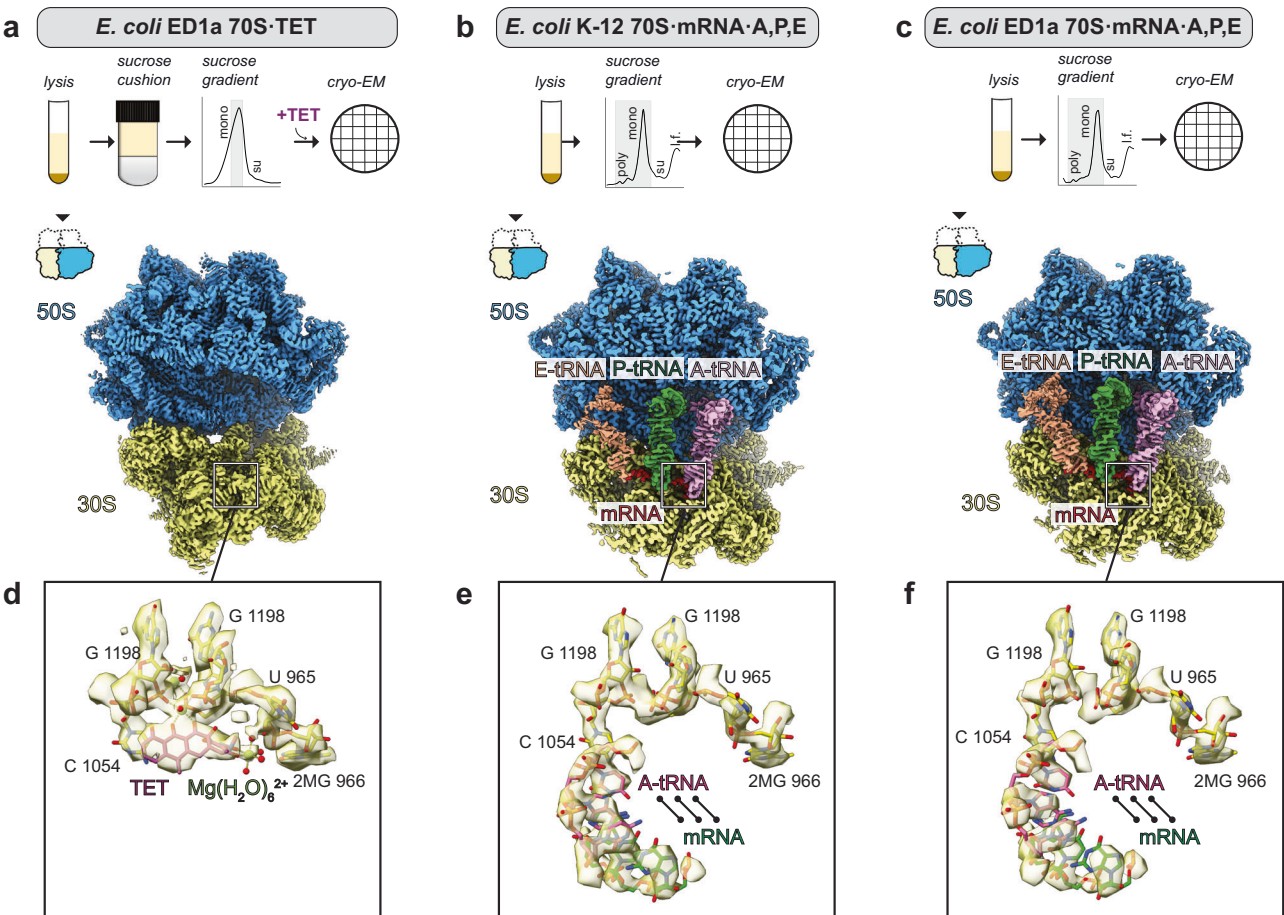

**Fig. 3 | Tetracycline-binding pocket is identical in ribosomes of both strains.**
**a** Single particle reconstruction of vacant 70S·TET complex from *E. coli* ED1a obtained from ribosome purification with sucrose cushion at 500 mM KCl. The maps of large subunit, and the small subunit head and the body were obtained individually by focused refinement. The schematic purification procedure is shown above the structure: mono – monosome fractions, su – ribosomal subunits fractions. The gray rectangle marks the fractions used for cryo-EM data collection. **b** Single particle reconstruction of 70S·mRNA·A,P,E complex from *E. coli* K-12 obtained from crude ribosomal extract. The map represents a class containing mRNA (red) and A-, P-, and E-tRNAs (pink, green, and beige respectively). The schematic purification procedure is shown above the structure: poly – polysome fractions, mono – monosome fractions, su – ribosomal subunits fractions, l.f. – light material fractions. The gray rectangle marks the fractions used for cryo-EM data collection. **c** Single particle reconstruction of 70S·mRNA·A,P,E complex from *E. coli* ED1a obtained from crude ribosomal extract. The map represents a class containing mRNA (red) and A-, P-, and E-tRNAs (pink, green, and beige respectively). The schematic purification procedure is shown above the structure: poly – polysome

fractions, mono – monosome fractions, su – ribosomal subunits fractions, l.f. – light material fractions. The gray rectangle marks the fractions used for cryo-EM data collection. **d** The TET binding pocket bound to antibiotic visualized on *E. coli* ED1a 70S·TET structure. The 16S rRNA is colored yellow, TET is colored magenta. Atomic coordinates of 16S were retrieved from PDB ID 8CF1, displayed atoms were fitted as rigid body into a density of TET-binding pocket segmented from the 30S-head map, and manually refined in real space. **e** The TET binding visualized on *E. coli* K-12 70 S·mRNA·A,P,E structure. The 16S rRNA is colored yellow, TET is colored magenta. Atomic coordinates of 16S are the same as in (**d**), mRNA (green) and A-tRNA (pink) coordinates were retrieved from PDB ID 4V9A and displayed triplet was fitted as rigid body into corresponding density segmented from the full 70S·mRNA·A,P,E map. **f** The TET binding visualized on *E. coli* ED1a 70S·mRNA·A,P,E structure. The 16S rRNA is colored yellow, TET is colored magenta. Atomic coordinates of 16S are the same as in (**d**), mRNA (green) and A-tRNA (pink) coordinates were retrieved from PDB ID 4V9A and displayed triplet was fitted as rigid body into corresponding density segmented from the full 70S·mRNA·A,P,E map.

strict purification and analyzed the crude ribosome mixture pooled from the sucrose gradient sedimentation (Fig. 3b, c), and in silico extracted 70S particles containing an A-tRNA (Supplementary Fig. 1b), as the engagement of the A-site with the tRNA conveys structural stability to the TET-binding region. We resolved the structures of 70S·mRNA·A,P,E complexes from both strains at 3.0 Å resolution (Fig. 3b, c) and showed that the nucleotides of the 16S rRNA bound to A-tRNA had identical arrangements in both strains (Fig. 3d−f). This region is also identical to the structures of 70S·TET complexes from other bacteria (Supplementary Fig. 1d−f). These results strongly support the notion that the binding mode of TET to the major site at the ribosome is the same in both strains and highly conserved across bacteria.

## TPP identifies ribosomal proteins and ribosome biogenesis factors as differentially affected

To investigate further why the two strains respond differently to TET, we applied the 2D-TPP approach[19] (Supplementary Fig. 2a). This method quantifies changes in protein abundance, thermal stability, and solubility, thereby creating a holistic overview of the proteome state and revealing potential off-target effects[20]. Protein stability is a good proxy for the binding capacities of proteins for small molecules that stabilize the respective binding pockets. For both strains, we collected TPP data after 30 min or 3 h exposure to TET (8 μg/ml, 2X MIC) and from untreated cells. The proteomic screen quantified more than 1800 proteins, 1219 of which overlapped in both strains, while 318 were *E. coli* K-12-specific, and 332 were annotated as *E. coli*

ED1a-specific (Fig. 4a−c, Supplementary Data 1, Supplementary Fig. 2b −f). Overall, after 30 min of TET exposure, changes in protein thermal stability were more pronounced than changes in abundance (Fig. 4a block T2, Supplementary Fig. 2c block T2), while prolonged TET treatment resulted in more prominent changes in protein abundance (Fig. 4a block T4, Supplementary Fig. 2c block T4). Thus, the earlier timepoint likely reflects a more immediate modulation of the thermal stability of available proteins, while cells reshape the proteome in the longer term, particularly untreated cells after 3 h, as they undergo a transition to the stationary phase (Supplementary Fig. 2c block T3). Thus, we focused our comparative analysis mostly on the short-term response (30 min), while timepoint 3 was excluded from a further comparative analysis.

Both strains showed a significant increase in abundance of cold-shock proteins, including RbfA, CspB, and CspG (Fig. 4b). Earlier, similar effects were reported for TET-treated E. coli K-12 by 2D electrophoresis[21] indicating that our approach and results are in line with prior biochemical studies. At the same time, we detect notable differences between the strains in abundance or stability scores for a number of proteins. Among them, the proteins of malate metabolism (MalE, MalM), and galactose metabolism (GatA, GatC, gatD, GatZ), and several translation-associated proteins including EttA, TrmA, RimO, translation initiation factor IF-1 (Fig. 5a). Noticeably, we detect strong changes in abundance or thermal stability over the course of TET treatment (T2 vs T4) for several proteins annotated only in one of the strains, including several uncharacterized proteins (Supplementary Fig. 2e). We designate these hits as strain-specific proteins.

Gene ontology term analysis identified ribosomal proteins (r-proteins), in particular components of the 50S large ribosomal sub-unit, as the major category affected by TET in terms of thermal stability (Supplementary Data 1, Fig. 4c, Supplementary Fig. 2f). Additionally, ribosome biogenesis factors, such as RimM, RimO, EttA displayed a significant increase of thermal stability scores in TET-treated ED1a cells (Fig. 5a). Notably, these, as well as r-proteins themselves, only showed high thermal stability fluctuations, whereas the abundance remained mostly unchanged over time and regardless of antibiotic treatment (Fig. 5a, b, Supplementary Fig. 3a−c). This suggests that the number of ribosomes remained constant in all conditions, but that their structural status was altered shortly after treatment, specifically for E. coli ED1a (Fig. 5b, Supplementary Fig. 3b). Thermal stability scores increased particularly for proteins near the subunit interface, an effect that may be attributed to reduced inter-subunit rotation (Supplementary Fig. 3b). Due to the high genetic similarity of both strains, this con-formational flexibility is unlikely to cause the drug sensitivity of ED1 cells, but rather reflects the partially compromised translation as a result of drug treatment. Overall, these data identify stress response, metabolic and translation factors to be affected by treatment, and underscore a stronger impact of the drug in ED1a cells.

## Cryo-ET visualizes distortion of the bacterial envelope in the gut strain

Recent studies demonstrated the power of subtomogram averaging to determine high-resolution structures of ribosomes and to resolve their different conformations in situ[3–5]. We performed in situ cryo-ET cou-pled to subtomogram averaging (STA) to visualize cell morphology and the structural state of the protein synthesis apparatus in the two E. coli strains upon TET treatment. We obtained lamellae using the focused ion beam (FIB) milling technique and collected the tilt series of the thin sections of individual bacterial cells at 30 min without treat-ment (30 min LB), and treated (30 min TET) (Fig. 6a−c).

Visual inspection revealed that TET-treated E. coli ED1a cells had a distorted cell envelope, in particular wobbling of outer and inner membranes (Fig. 6d, e). We measured the periplasmic thickness of the cells using computational segmentation of membranes and calculated the distance between inner and outer membranes as previously

described[22]. Upon treatment with TET, the uniformity of the peri-plasmic space of E. coli ED1a cells was significantly perturbed, which resulted in overall longer intermembrane distances across the cells and irregularity within individual cells (Fig. 6f; Supplementary Fig. 4a −e). Although the colony counting assay showed that at least half of ED1a population was still viable after 30 min of TET treatment (Fig. 2c), the envelope disruptions were visualized in virtually every cell ana-lysed by cryo-ET, suggesting that the damage of some cells was still below a threshold allowing recovery. This is in accord with the TPP data that had identified porins OmpF, OmpC; multidrug efflux pumps AcrAB-TolC, MdtE, EmrA; ABC (ATP-binding cassette) transporters Dpp, Opp, MsbA, Mal, Rbs; and cell envelope-shaping proteins, OmpA, TolR to differ in abundance and thermal stability between both strains (Supplementary Data 1, Supplementary Fig. 2d).

## The majority of ribosomes in the gut strain are in a translation incompetent state upon TET treatment

To obtain spatial information on the 70S ribosomes in untreated and treated cells, we applied the STA approach. It is needed to note that we did not pre-select the cells used for cryo-ET data collection, inter-membrane distances measurements, and STA. Considering the high degree of structural similarity observed in our cryo-EM structures of ribosomes in the two strains, we merged 70S subtomograms from all datasets to obtain a resolution benchmark for our STA pipeline (Fig. 7a −d, Supplementary Fig. 5a, b). The resulting subtomogram average of E. coli 70S ribosomes overall reached a resolution of 5.2 Å (Fig. 7c, Supplementary Fig. 5a). Some regions of the structure were resolved to side-chain details. Among them, Lys44 and Arg46 of the protein uS4 that supports placement of mRNA at the mRNA entry pore formed by uS3, uS4, and uS5, were defined at this depth (Fig. 7d).

We then determined 70S ribosome structures from E. coli K-12 and E. coli ED1a for control (30 min LB) and 30 min TET-treated cells indi-vidually at a resolution range from 7.0 to 8.5 Å (Fig. 7e−h). Even though at this resolution range the drug density remained undistinguishable, it was possible to disentangle the different functional states of ribo-somes in each of the four conditions, which yielded striking differ-ences between the two strains upon TET treatment. We performed multiple rounds of 3D classification and sampled ribosome structure heterogeneity based on ligand occupancy (Fig. 7i−l, Supplementary Fig. 6a−d). Without the antibiotic, we observed the densities attributed to mRNA, P-site tRNA and the nascent peptide similarly for both strains (Fig. 7e, f). The most abundant classes in untreated samples repre-sented 70 S·EF-Tu·A/T,P,E complex and 70 S·EF-G·P,E complex (Fig. 7i,j Supplementary Fig. 6a, b), resembling the slow translation states of tRNA accommodation and peptide bond formation[23–25]. For the K-12 30 min TET sample, we observed similar ligand composition (Fig. 7g), and all classes contained P-tRNA and elongation factors, suggesting that these ribosomes are translation competent (Fig. 7k, Supplemen-tary Fig. 6c). In striking contrast, the densities for the mRNA, P-tRNA, and the nascent chain were absent in 70S of E. coli ED1a 30 min TET (Fig. 7h), implying differences in the translation capacity. Subse-quently, 3D classification revealed that 96% of TET-treated ED1a ribo-somes do not carry P-tRNA, a ribosomal state that is incompatible with protein synthesis. A large proportion of ribosomes in this sample were annotated as 70S·EF-Tu·A/T,E complexes (Fig. 7l, Supplementary Fig. 6d). Canonically, a complex of EF-Tu with aminoacyl-tRNA (aa-tRNA) binds to the ribosome to promote the elongation of the nascent chain bound to the P-tRNA[1]. The observed presence of EF-Tu on the P-tRNA-deficient 70S could potentially be caused by the generation of run-off ribosomes that could not be recycled for translation initiation.

Interestingly, regardless of TET treatment 70S, averages con-tained density for the ribosomal protein bS1, with the resolved N-terminal ribosome anchor and the domain I regions (Supplementary Fig. 6e). This suggests that after aiding mRNA accommodation on the 30S subunit[26,27], bS1 remained bound to the ribosome. This finding

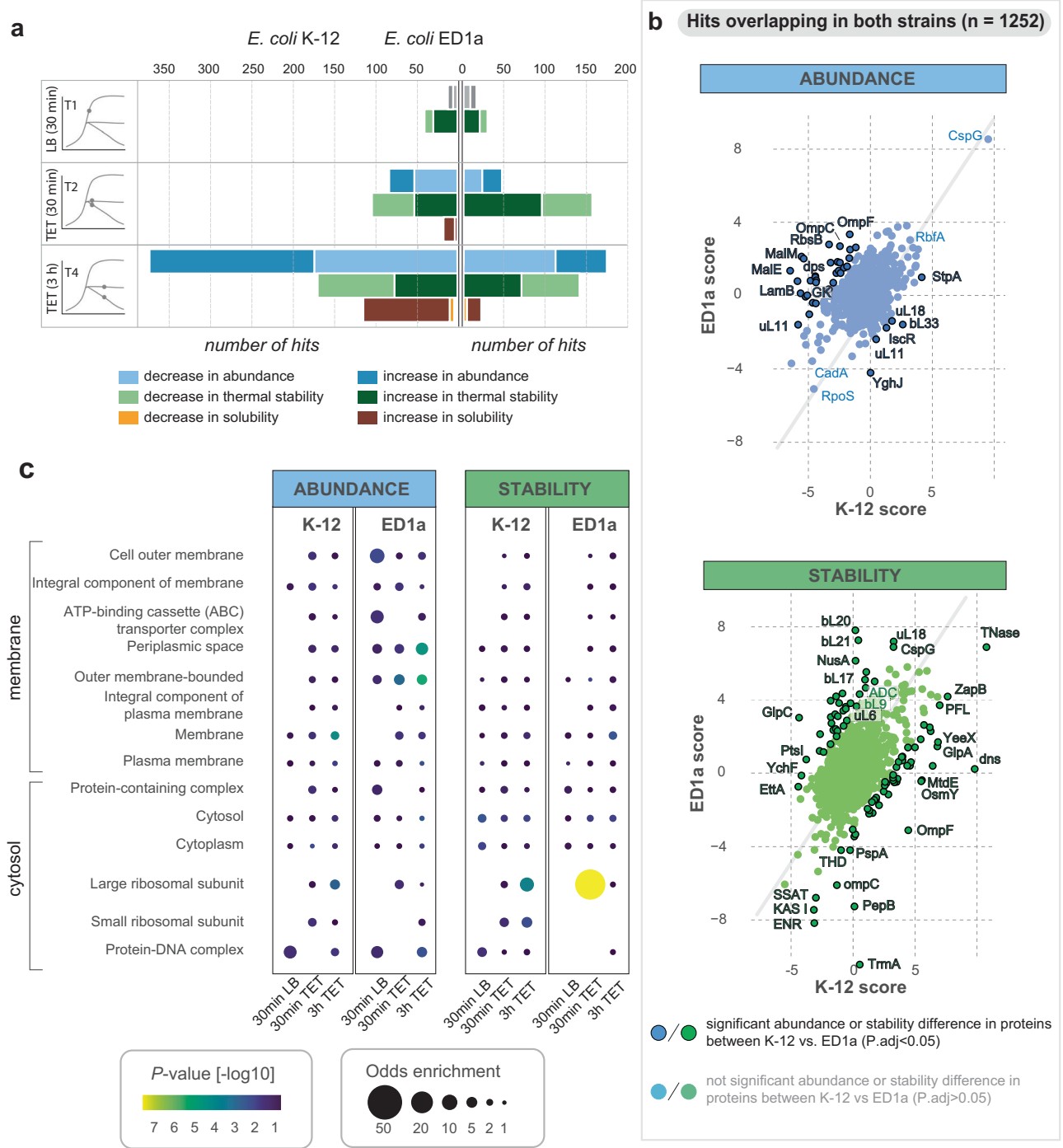

**Fig. 4 | 2D-TPP identifies ribosomal proteins and ribosome biogenesis factors as differently affected by TET in the two strains. a** Comparison of the number of significantly affected proteins in abundance (blue shades), thermal stability (green shades), and solubility (orange/brown shades) for *E. coli* K-12 (left side) and *E. coli* ED1a (right side). Timepoint T1 represents untreated cells grown for 30 min, timepoints T2 and T4 represent TET-treated cells grown for 30 min and 3 h, respectively. The scores are calculated relative to the control condition T0 (untreated cells, 0 min LB). Timepoint T3 (3 h, untreated control) is shown in Supplemental Fig. 2C since the data were not comparable as cells shifted to stationary phase only in this set-up. **b** Scatter plot of abundance (upper panel) and stability values (lower panel) for TTP hits overlapping in both strains for T2 (30 min TET), comparing K-12 scores (*x*-axis) against ED1a scores (*y*-axis). A two-sided *t*-test was applied based on the effect size distribution resulting from the respective K-12 vs. ED1a scores. *P* values were adjusted using the Benjamini-Hochberg method. Each dot represents a protein; if it is highlighted with a black contour, it indicates that there is a significant difference between K-12 and ED1a scores ($P_{adj} < 0.05$, confidence interval 0.95). Exact *p* values for each protein can be found in Supplementary Data 1. The color scheme is explained in the legend below the figure panel. **c** Gene ontology (GO) enrichment plot of TPP results for different cellular compartments, with compartments ordered according to their physical location on the outer membrane – cytosol axis. The enrichment for each GO-term in the significant protein set (global & local FDR < 0.05) in each respective condition (30 min LB, 30 min TET, 3 h TET) is calculated using the Fisher Exact Test (two-sided) relative to the insignificant portion of proteins in each condition (displayed as color gradient for p from highest (yellow) to lowest (blue) significance). If the GO-term is enriched in at least one condition (*P* < 0.01) and has at least 10 significant protein components significantly affected in the dataset, it is shown on the *y*-axis of the bubble plot. The bubble size indicates the odds enrichment, whereas the bubble color reflects the *P* value (-log10 scale).

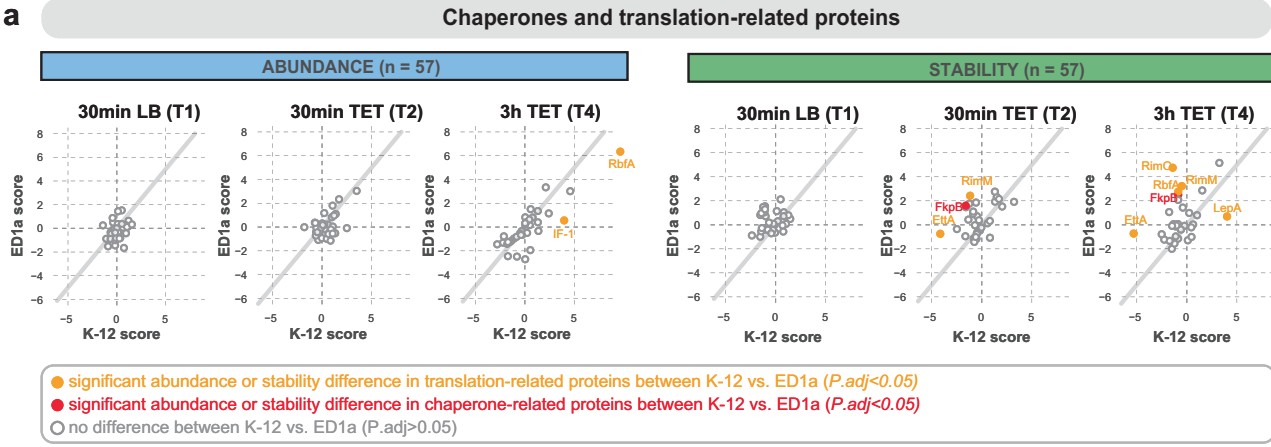

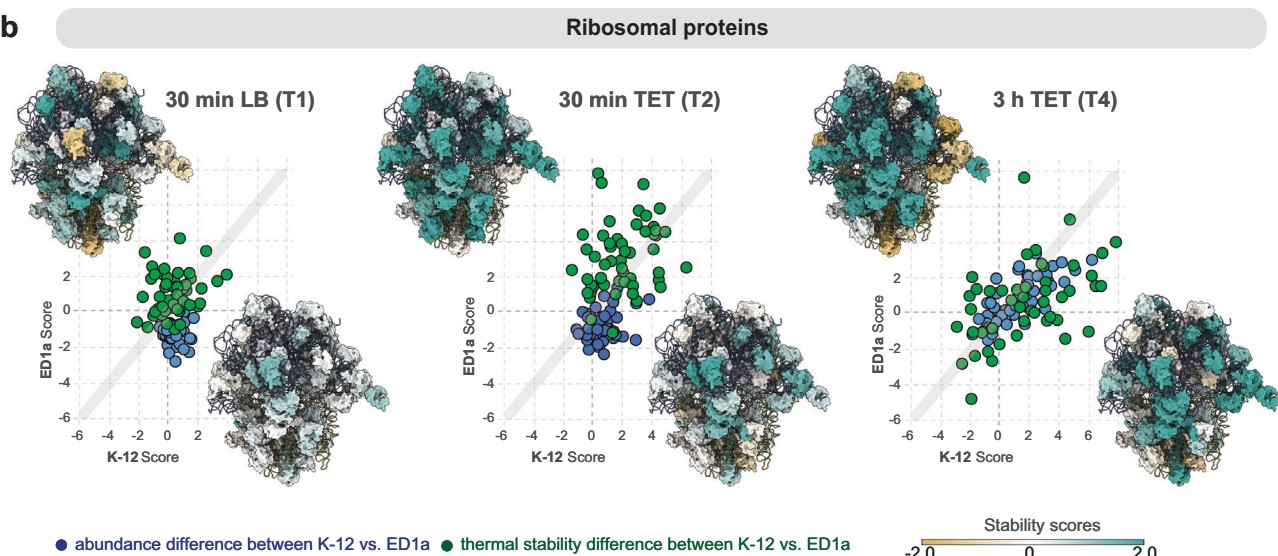

**Fig. 5 | Thermal stability of translation-related and ribosomal proteins is affected stronger than their abundance. a** Scatter plot of abundance and stability values for proteins involved in translation and chaperoning functions for T1 (30 min LB), T2 (30 min TET), and T4 (3 h TET) respectively, comparing K-12 scores (*x*-axis) against ED1a scores (*y*-axis). A two-sided *t*-test was applied based on the effect size distribution resulting from the respective K-12 vs. ED1a scores. *P* values were adjusted using the Benjamini-Hochberg method. Each dot represents a protein; orange and red coloring indicates a significant difference between K-12 and ED1a scores ($P_{adj} < 0.05$, confidence interval 0.95). Exact p values for each protein can be found in Supplementary Data 1. **b** Scatter plot of abundance (blue) and stability values (green) for ribosomal proteins for T1 (30 min LB), T2 (30 min TET), and T4 (3 h TET) comparing K-12 scores (*x*-axis) against ED1a scores (*y*-axis). Each dot represents a protein. The two insets in each plot depict 70 S ribosome structures shaded according to the respective thermal stability score values of the respective protein components (upper left corner: coloring based on ED1a scores; lower right corner: coloring based on K-12 scores). The PDB model was combined from coordinates fetched from PDB 3J7Z, PDB 7K00, and PDB 6H4N. Proteins with increased stability are colored turquoise, proteins with decreased stability are colored tan, and proteins with unchanged stability are colored white.

corroborates the vital role of this protein in *E. coli* translation[28] and indicates that binding may be required for the later stages of the translation cycle, transcription-translation coupling[29], or ribosome hibernation[30]. Overall, these results suggest that after 30 min TET treatment *E. coli* ED1a contains a very small pool of ribosomes that are potentially capable of protein synthesis.

## Discussion

Tetracyclines are by and large considered bacteriostatic antibiotics, yet the two *E. coli* strains analyzed in our study exhibit differential responses, with bactericidal activity seen in ED1a (Fig. 2c, f)[8]. It has previously been postulated that ribosome-binding antibiotics can be bactericidal for cells with few ribosomal operons, which also have reduced growth rates[31]. However, considering that both *E. coli* K-12 and *E. coli* ED1a have seven ribosomal operons and a comparable growth rate (Fig. 2a), we concluded that the killing of the gut isolate

by TET is unlikely to be linked to altered rRNA transcription. Moreover, the number of ribosomes per cell detected and resolved by cryo-ET is comparable in both strains (Supplementary Fig. 5b), regardless of presence of the antibiotic. In addition, the effect of increased intracellular concentration of antibiotic on *E. coli* survival appears to be complex and may depend on numerous factors, as we observed that *E. coli* K-12 retains the bacteriostatic phenotype even upon treatment with 16 µg/ml TET, a concentration which results in a similar intracellular concentration to ED1a treated with 8 µg/ml (Fig. 2d−f). Although our analysis does not differentiate if this is a direct effect or is an indirect consequence of effects such as increased drug degradation in the lab strain, we could rule out the possibility of differential interaction of the antibiotic with the ribosome. Direct binding of TET to the 30S A-site of the bacterial ribosome is well documented[12,13]. Here, we show that ED1a and K-12 70S ribosomes undergo an indistinguishable interaction with the

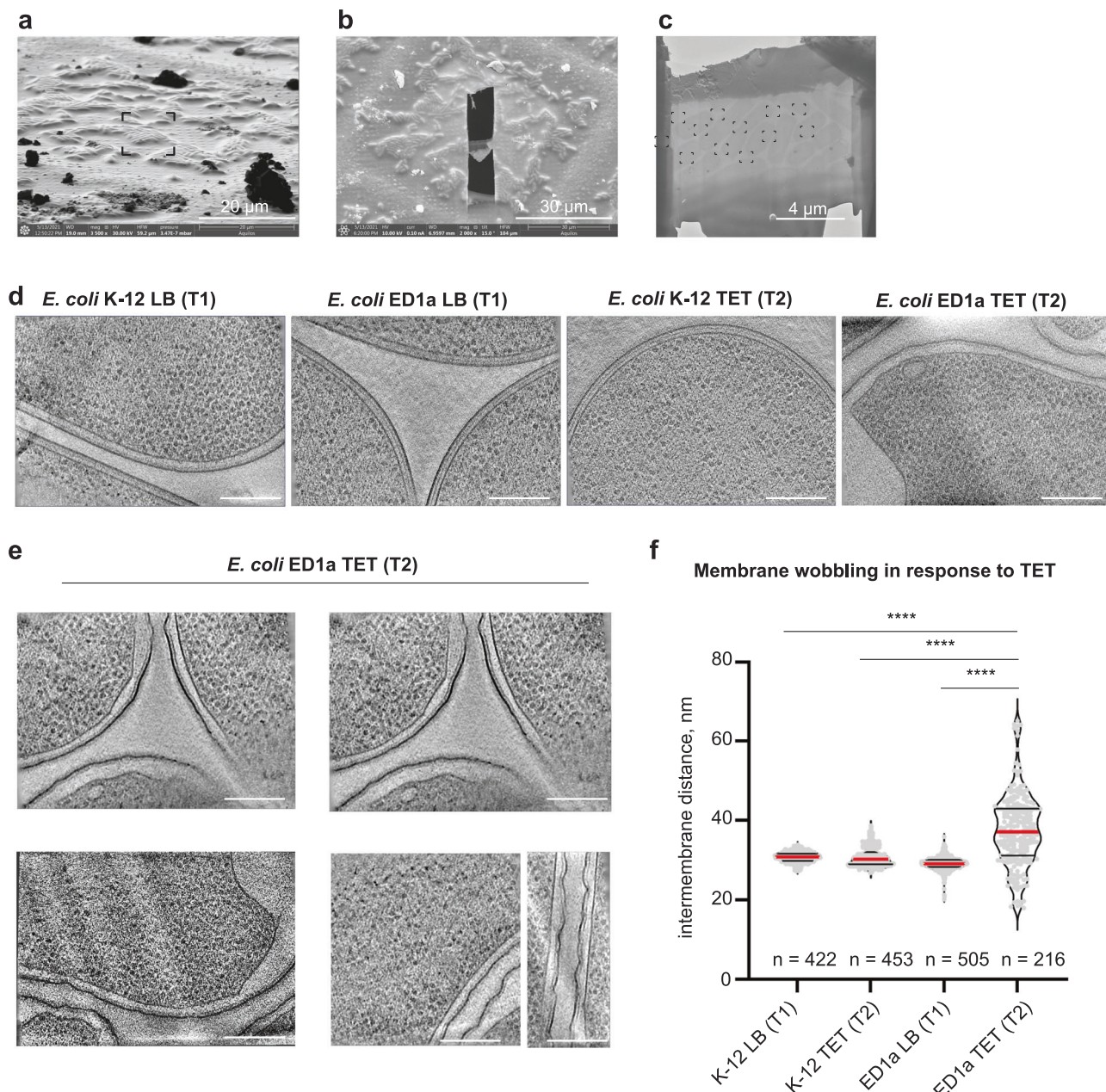

**Fig. 6 | Cryo-ET reveals severe membrane wobbling in *E. coli* ED1a cells treated with TET. a** Ion beam image of the *E. coli* cells selected for FIB-milling. Scale bar 20 μm. For each strain, cell batches for FIB milling were prepared in three biological replicates from independent cell cultures grown from individual colonies. Cryo-EM grids for each sample were prepared in at least two technical replicates. **b** Scanning electron beam image of the grid square with FIB-milled lamella of bacterial cells. Scale bar 30 μm. For each cryo EM grid, at least three lamellae were prepared, comprising 5–15 individual cells per lamella. **c** Transmission electron microscopy (TEM) image map of a lamella. The black targets point to the regions of subsequent tilt series data collection. Scale bar 4 μm. For each obtained lamella, maximum number of tilt series were collected, excluding pre-selection of cells. **d** Digital slices from the reconstructed tomograms, demonstrating the membranes wobbling in *E. coli* ED1a cells treated with 8 μg/ml (2X MIC) TET for 30 min (right-most panel).

Scale bar 0.2 μm. T1: timepoint 1 (30 min LB), T2: timepoint 2 (30 min TET). **e** Digital slices at different z-heights from tomograms of various *E. coli* ED1a cells treated with TET for 30 min. The images were low pass filtered to better visualize the membrane wobbling. Scale bars 0.2 μm. **f** Intermembrane distance in the cells used for cryo-ET and subtomogram averaging. The violin plot shows the smoothed density of every measurement (gray dots), the first and third quartiles (black solid lines), and the median (red solid line); *n* represents the number of points at which measurements were made for 10 K-12 (LB), 10 K-12 (TET), 10 ED1a (LB), and 7 ED1a (TET) cells. The violin plots for each of these cells individually is shown in Supplementary Fig. 4a–d. The schematic cartoon of intermembrane distance analysis is shown in Supplementary Fig. 4e. The statistical significance was calculated using a two-sided unpaired *t* test. The significance symbols indicate *P* values: ns for *P* = 0.6758 **** for *P* < 0.001. Source data are provided as a Source Data file.

antibiotic (Fig. 3d–f, Supplementary Fig. 1c–f). Together these findings demonstrate that the effect of TET on these bacterial cells is more complex and that multiple aspects likely have to be considered to explain the overall phenotypic differences observed.

In addition to pointing towards an involvement of ribosomal biogenesis, translation, and protein folding (Fig. 4c, Fig. 5a, b,

Supplementary Fig. 3a–c), we detected subtle differences in the proteomic makeup of membrane proteins, including proteins that have been previously associated with antibiotic response (Supplementary Data 1, Supplementary Fig. 2d). Previous proteomic and mutagenesis/genetic analyses of *E. coli* K-12, *Acinetobacter baumannii* DU202, and an environmental-borne *E. coli* EcAmb278 also found an alteration

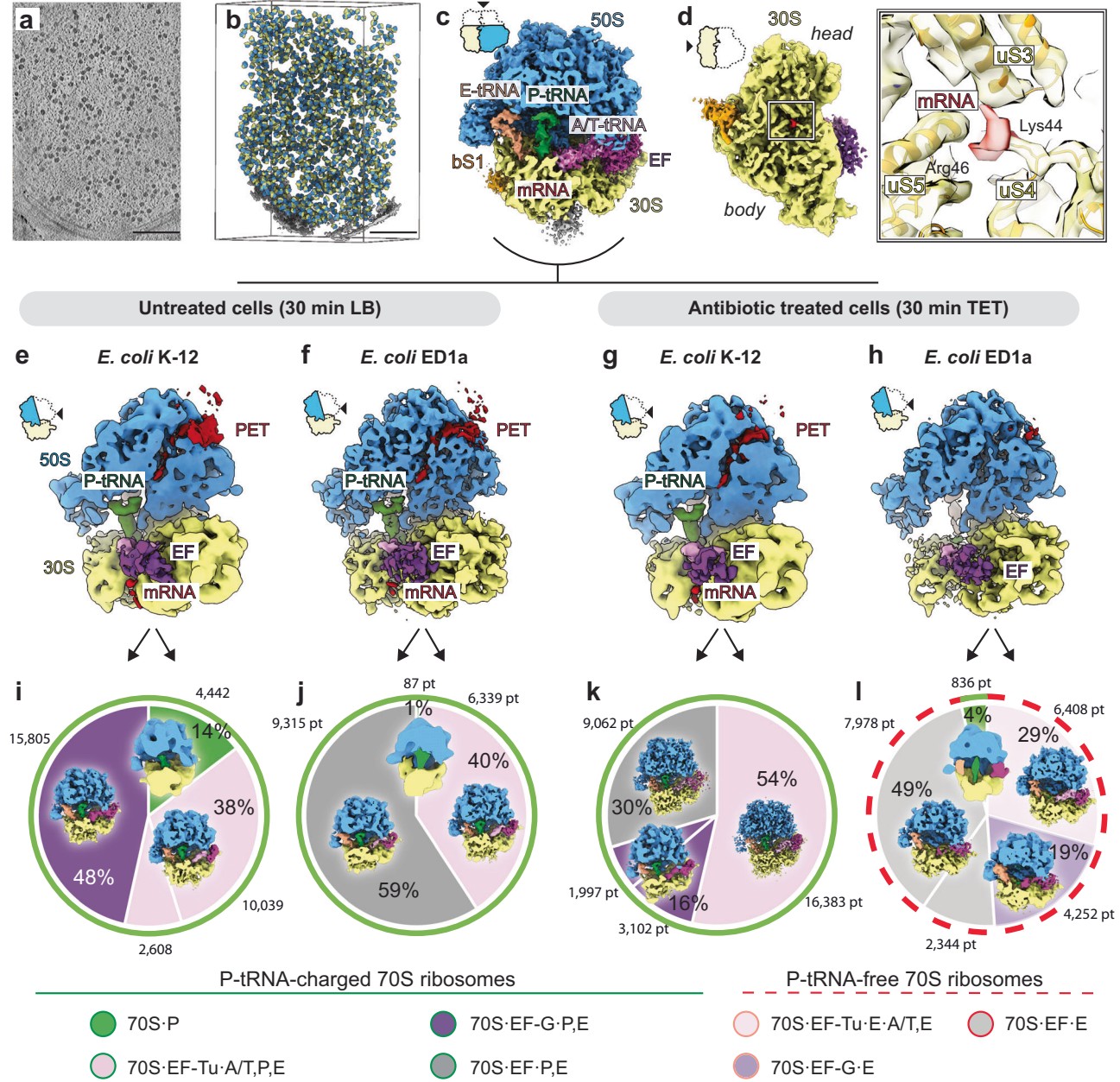

**Fig. 7 | In situ cryo-ET, subtomogram averaging and 3D classification reveals that the majority of *E.coli* ED1a 70S ribosomes are in translation-incompetent state lacking P-tRNA after TET treatment. a** Digital slice from a representative tomogram of *E. coli* ED1a cells. The tomogram was denoised and the contrast was enhanced using IsoNet[59] trained on 5 tomograms with different defocus. Ribosomes appear as dark spherical objects. Scale bars 0.2 μm. **b** Three-dimensional view of the tomogram from (**a**) with aligned ribosome particles projected back into tomographic volumes. Small ribosomal subunits colored yellow, large subunits colored blue. Membranes segmented from tomogram are colored in gray. **c** The subtomogram average of the *E. coli* 70S ribosome from all datasets was resolved at 5.3 Å resolution. The structure represents an actively translating ribosome with mRNA (red), A-, P-, E-tRNA ligands, (pink, green, and beige respectively), elongation factors (EF, magenta) bound. **d** The solvent side view of the small ribosomal subunit of the overall subtomogram average of *E. coli* 70S ribosome. The enlarged inset shows details of the mRNA entry tunnel composed of universal r-proteins uS3, uS4, and uS5. The mRNA is supported by Lys44, and Arg46 of uS4, for which density is resolved to side-chain resolution. The segmented density for r-proteins shown in yellow, for mRNA in red. For visualization reasons, and to make the mRNA entry tunnel visible, the extra density at the polysome interface between the head and the body of the 30S is omitted. It can be seen colored in gray in figure (**c**).

Subtomogram averages of the 70 S ribosome from (**e**) – *E. coli* K-12 (LB) at 8.5 Å, (**f**) – *E. coli* ED1a (LB) at 7.5 Å, (**g**) – *E. coli* K-12 (TET) at 8.5 Å, (**h**) – *E. coli* ED1a (TET) at 7.0 Å. The structure from the *E. coli* ED1a TET sample (**h**) shows only negligible density for tRNA and peptide exit tunnel (PET), suggesting the translation-incompetent state of most of the ribosomes in these cells. The 50S subunit is clipped to visualize the nascent chain inside PET (dark red), and the P-tRNA (green). Ribosomal subunits and ligands coloring: 30S – yellow, 50S – blue, mRNA – red, A-, P-, E-tRNA ligands – pink, green, and beige respectively, elongation factors (EF) – magenta. The distribution of ribosome classes sampled from *E. coli* K-12 30 min LB (**i**), *E. coli* K-12 30 min TET (**j**), *E. coli* ED1a 30 min LB (**k**), *E. coli* ED1a 30 min TET (**l**). Each segment of the pie chart represents a separate class obtained by 3D classification: 70S·P complex – green, 70S·EF-Tu·A/T,P,E complex – pink, 70S·EF-G·P,E complex – purple, 70S·P,E complex with weakly resolved elongation factor (EF) – gray, 70S·EF-Tu·A/T,E complex – light pink, 70S·EF-G·E complex – orchid, 70S·E complex with weakly resolved elongation factor (EF) – light gray. The thick green contour defines ribosome classes of translation-competent ribosomes charged with P-tRNA, the thick red dashed contour defines classes of translation incompetent ribosomes lacking P-tRNA. The thin black contours outline structurally similar 3D classes. Number of particles for each class is shown next to corresponding sector. Source data are provided as a Source Data file.

in the abundance of several membrane transporters, including OmpC, TolC, LamB and peptidoglycan-based cell wall proteins upon TET treatment[32–35]. The differential effects we detected in the two strains by proteomics could be related to the phenotypic differences observed in membrane morphology by cryo-ET (Fig. 6d, e). It is unlikely that such a severe envelope distortion is an effect only of blotting during cryo-ET sample preparation, as mechanical forces would presumably affect cells of both strains. Noteworthy, based on observations of membrane wobbling, TET and its analogs were previously suggested to have an additional mode of action, independent of translation inhibition[36,37]. In our large-scale 2D-TPP screen, we identify numerous species-specific proteins and putative uncharacterized proteins with significant changes in their abundance and thermal stability upon TET treatment (Supplementary Data 1, Supplementary Fig. 2e). These include several proteins with yet unknown or poorly described function, where a potential role in the response to TET would need to be further assessed.

The ability of cells to cope with stress conditions and to quickly recover, strongly depends on the activity of the translation machinery, and the functional state of the ribosomes[38]. Using cryo-ET, we find that in contrast to *E. coli* K-12, in *E. coli* ED1a cells, TET treatment leads to presence of almost all ribosomes (96%) in a very specific state of 70S containing translation factors and E-tRNA, but lacking the P-tRNA (Fig. 7l, Supplementary Fig. 6d). This conformational state is incompatible with translation, as the P-tRNA, which retains the nascent peptide chain on the ribosome, is required for every round of incorporation of a new amino acid. This state also does not seem to be a result of ribosome hibernation. In our cryo-ET data, we did not observe any class of ribosomes inactivated by stress proteins, such as 50S·RsfS complexes[39], or 100S and 70S complexes[40,41], which are mediated by hibernation promoting factors like HPF, or recently characterized Balon protein[42]. We also did not detect significant changes in abundance or thermal stability of HPF by proteomics (Supplementary Data 1). Despite the fact that the HPF binding site is known to overlap with that of TET[43] and thus could theoretically protect ribosomes by competitive binding, our results indicate that upon TET treatment, *E. coli* ribosomes are preferentially retained in the non-hibernating 70S form. This is true even in the ED1a strain, which experiences a dramatic conversion of nearly all ribosomes to a translation-incompetent state, which might be irreversible or require considerable amount of energy to overcome.

In contrast, P-tRNA is present in every class of the K-12 TET sample, indicating that these ribosomes likely remain translation competent, and possibly restart protein synthesis upon TET removal. This could then be a very rapid response, as our cryo-ET structures show that in *E. coli* K-12 cells, the EF-Tu·aa-tRNA complex can bind to the 70S ribosome even in the presence of TET in the medium (Fig. 7g, Supplementary Fig. 6c). This suggests a high competition between reversible binding of antibiotic and tRNA to the ribosomal A-site inside cells. Unfortunately, the A-tRNA-containing state of the ribosome could not be resolved into a separate class in our in situ data, while the 70S·mRNA·A,P,E PRE-translocation state was one of the most abundant in the purified sample (Supplementary Fig. 1b). We reason that this is likely due to a more transient nature of this state within a cellular environment in presence of translation factors and cofactors, as well as the technical limitation of the in situ data.

In the recent in situ cryo-ET studies of *Mycoplasma pneumonia* bacterium treated with other ribosome-binding antibiotics such as chloramphenicol or spectinomycin, the P-tRNA-lacking state of the ribosome was not observed[4,44]. A similar approach in untreated eukaryotic cells found a fraction of ribosomes with the vacant P-tRNA site, but determined these as being in a hibernating-like state[3,5]. Moreover, in human cells treated with translation inhibitor homoharringtonine, only 47 % of ribosomes turned into P-tRNA-deficient state, despite showing the drug bound to the majority of ribosome

particles[5]. All this suggests that the translation-incompetent state observed in ED1a-TET sample is different from inactive ribosomes previously observed in eukaryotic cells. The response of *E. coli* ED1a to TET is complex and may not affect exclusively translation regulation but also manifests itself in various phenotypes such as membrane distortions and an increase in intracellular antibiotic concentration. Whereas some of these phenotypes may be `triggered not by the antibiotic itself but by cell death or other factors which have yet to be discovered, it is plausible to assume that the redistribution of translation states in the cells is a direct consequence of antibiotic action on the ribosome, as its main target in the cell.

From a technical perspective, most of bacterial species, including *E. coli*, are too thick for the direct imaging by transmission electron microscopy, and require thinning by focused ion beam, similar to any eukaryotic cell. Moreover, the dense molecular crowding in the cytosol and the absence of cellular compartmentalization make *E. coli* a challenging sample for cryo-ET tilt-series alignment and high-resolution subtomogram averaging. Such limitations likely restricted many cryo-ET studies on bacteria to the analysis of phenotypic features, or structure determination of peripheral macromolecular complexes (reviewed in ref. 45). On top of that, quantitative structural analysis of highly dynamic molecular machineries such as the ribosome require a very delicate sample preparation with no sample intervention before plunge-freezing. In this study, we established a protocol for close-to-native *E. coli* sample preparation for cryo-ET (Methods) suitable for high-resolution subtomogram averaging of cytosolic complexes and thereby provide high-resolution in situ snapshots of ribosomes and the structural malfunctions induced by TET both in lab *E. coli* strains and in human gut isolate.

Our study shows that in situ and in-depth molecular analyses are essential to further advance the overall understanding of antibiotic susceptibility, in particular of human commensal bacteria that may be differentially affected by antibiotic treatment. Given the increasing evidence for a vital role of the microbiota in the maintenance of human health[46], preserving its homeostasis and minimizing interference with its species diversity is highly important, and the development of efficient species-specific antibiotics with reduced side effects represents one of the main challenges.

## Methods

### Growth and viability assays of E. coli treated with TET at 2X MIC

For all experiments, we used the standard lab strain *E. coli* BW25113 (K-12) and a gut strain *E. coli* ED1a available in the collection of Dr. A. Typas lab (EMBL-Heidelberg). Cells were grown in 250 ml flasks containing 50 ml LB at 37 °C and 180 rpm shaking. For growth curve estimations, the cells were grown continuously for 26 h. For the viability assays, 50 ml of cell cultures were inoculated with the overnight pre-culture to the starting $OD_{600}$ of 0.005, and grown for 3 h in LB until $OD_{600}$ of 0.3, then 20 ml of these cultures were transferred to new flasks containing 20 ml of prewarmed LB or LB supplemented with 16 µg/ml tetracycline (TET). These dilutions were made to prevent the transition of control LB-grown cells to the stationary phase during the first 30 min. The cultures were further grown for 30 min and 3 h. At every timepoint, 1 ml of culture was taken for OD measurements, and 1 ml for colony forming units (CFU) estimation. To address cell viability (CFU/ml) at 8 µg/ml TET, 1 ml of cultures were washed twice in 1 ml of phosphate buffer saline (PBS) and subjected to serial dilutions in PBS. The drops of 5 µl were placed on the LB-agar plates and incubated overnight at 37 °C. To estimate viability at different TET concentrations, cell cultures of 10 mL were grown in 15 ml tubes. At every timepoint, 0.5 ml of cultures were taken for OD measurements and CFU estimation. Cells were washed twice in PBS and plated on a solid LB-agar medium as described above.

## Intracellular TET measurements

Cell cultures of 50 ml were inoculated with the overnight pre-culture to the starting $OD_{600}$ of 0.005, and grown for 3 h in LB until $OD_{600}$ of 0.3. Then, 20 ml of cultures were diluted twice in prewarmed LB or LB supplemented with 16 µg/ml tetracycline (TET). After an additional 30 min of incubation, the cells were collected by centrifugation at $4000 \times g$ and 4 °C for 10 min. Cells were washed twice in 2 ml of PBS, suspended in 100 µl of PBS, and lysed by heating at 100 °C for 5 min. To check antibiotic thermal stability, TET dissolved in PBS was also heated for 5 min at 100 °C and its absorbance spectra of serial dilutions were measured in NanoDrop device (Thermo Scientific). To measure intracellular antibiotic concentration in cultures treated with different initial TET concentrations (from 0 to 64 µg/ml), the 50 ml cultures grown in 250 ml flasks for 3 h in LB were split into 5 aliquots of 10 ml and diluted twice with LB supplemented with an appropriate amount of TET. The cultures were further incubated for 30 min and lysed as described above.

## Ribosome purification

Both strains, *E. coli* K-12 and *E. coli* ED1a were grown for 3 h until reaching $OD_{600}$ of 0.3 in 1 L flasks filled with 200 ml of LB. Cells were collected in 400 ml centrifuge tubes filled with 150 g of frozen buffer A (10 mM HEPES-KOH pH 7.5, 100 mM $NH_4Cl$, 15 mM Mg-acetate, 1 mM ethylenediaminetetraacetic acid (EDTA), 1 mM DTT), and pelleted at $4000 \times g$ for 20 min at 4°. Pellets were resuspended in 7 ml of buffer A supplemented with the addition of 200 µl of protease inhibitor cocktail (one tablet (Roche), dissolved in 1 ml buffer A), of 50U DNase I (Roche) and lysed by three passages through French Press at 1000 Bar. Cell debris was removed by centrifugation at $30,000 \times g$ for 60 min.

For *E. coli* ED1a vacant 70S ribosome preparation, the supernatant was diluted to 6 ml in buffer A, layered onto 3 ml sucrose cushion prepared in high-salt buffer B (10 mM Hepes-KOH pH 7.5, 500 mM KCl, 15 mM Mg-acetate, 0.5 mM EDTA, 1 mM DTT), and spun at $100,000 \times g$ for 8 h at 4° in Ti 70.1 rotor (Beckman Coulter). The pellet was suspended in buffer E (10 mM Hepes-KOH pH 7.5, 60 mM KCl, 15 mM Mg-acetate, 0.5 mM EDTA, 1 mM DTT), and layered onto 5−30% sucrose gradients freshly prepared in buffer E supplemented with respective concentration of sucrose. The gradients were spun at $38,645 \times g$ for 15 h at 4° in SW28 rotor (Beckman Coulter). For total ribosomes preparation from both, K-12 and ED1a strains, the sucrose cushion step was by-passed, and lysates diluted to 1.5 mg/ml, and 0.2 ml were directly layered on 10−30% sucrose gradients prepared in buffer E, and spun at $32,600 \times g$ for 15 h at 4° in SW32 rotor (Beckman Coulter). For each sample, ribosome fractions were pooled as shown in Fig. 3a–c. The sucrose was removed and ribosomes were concentrated to 10 mg/ml using Amicon centrifugal filters with molecular weight cut-off 100 kDa. Aliquots of 10 µl were flash-frozen in liquid nitrogen, and stored at −80°.

## Single particle cryo-EM grids preparation and data collection

For ED1a 70S-TET dataset, vacant 70S ribosomes were mixed with TET to a final concentration of antibiotic of 0.2 mM and clarified by centrifugation at $10,000 \times g$ for 5 min. Samples of total ribosomes from K-12 and ED1a strains were clarified by centrifugation at $10,000 \times g$ for 5 min directly after thawing. For each sample, ribosomes (3 µl) were applied to Quantifoil R 1.2/1.3 300 mesh grids glow discharged for 90 s at 15 mA and 0.38 mbar using PELCO easiGlow. The sample was mounted on Vitrobot Mark IV (Thermo Scientific) at 100% humidity and 10 °C, then blotted for 3 s at nominal blot force 4 and plunge-frozen in liquid ethane, cooled by liquid nitrogen. Single particle cryo-EM data were collected using EPU software v2.14 (Thermo Scientific) on a 300 kV Titan Krios transmission electron microscope (Thermo Scientific) equipped with a BioQuantum-K3 imaging filter at the in-house electron microscopy facility of Max-Planck Institute of Biophysics. Dose-fractionated movies were acquired in electron counting

mode at 105,000 X magnification on a K3 camera, corresponding to a pixel size of 0.837 Å/pix. The calculated total electron exposure per image was 40 e⁻/Å² on the specimen and the exposure rate was set to 15 e⁻/pix/sec on the camera. A total of 12,022 images were collected for *E. coli* K-12 samples, and 16,559 images for *E. coli* ED1a samples. CryoSPARC Live[47] was used for on-the-fly data quality assessment.

## Single particle cryo-EM image processing and model fitting

Image processing was performed in CryoSPARC v3.3.1[47]. After patch motion correction and contrast transfer function (CTF) estimation, all micrographs were kept for further analysis. A total of 1,740,048/ 2,302,674 (*E. coli* K-12 / *E. coli* ED1a) particles were picked using the blob picker with minimum and maximum particle diameters of 200 Å and 300 Å respectively, and extracted using a box size of 64 pixels at 3.348 Å/pix. The 2D classification was used to remove junk particles resulting in 1,211,457/1,167,533 particles that were further used for further classification. Using 70S ribosome volume generated by ab initio reconstruction from 10,000 randomly selected particles, the heterogeneous refinement was used to remove poorly aligned volumes. The remaining particles were re-extracted with the box size of 200 pixels at 1.674 Å/pix. A new ab initio model was generated, and another round of heterogeneous refinement was performed to reveal conformational heterogeneity of ribosomes. Particles of the 70S·mRNA·APE-tRNA complexes (80,169/132,290 particles for K-12/ ED1a respectively) were extracted with a box size of 300 pixels at 1.256 Å/pix and subjected to homogeneous refinement with default parameters. The final resolution was 3.02 Å, and 3.01 Å for K-12 and ED1a respectively. An overview of the single particle cryo-EM processing workflow is shown in Supplementary Fig. 2. The model of *E. coli* 70S·mRNA·A,P,E-paromomycin (PDB ID: 7K00) complex was rigid-body fitted into obtained experimental density maps. The 16S rRNA nucleotides around the TET-binding region were manually refined using Coot[48]. The resulting model was aligned with *E. coli* 70S·TET complex from *E. coli* (PDB ID: 5J5B) and *Thermus thermophilus* (PDB ID: 4V9A) models in ChimeraX[49] using MatchMaker restricted to TET-Binding region. The summary of data collection parameters and image processing statistics is given in Supplementary Table 1.

## Thermal protein profiling and sample preparation for MS

Thermal proteome profiling was done as previously described[19] with adaptation of cell growth conditions to our current experimental setup. Cultures were inoculated with the overnight pre-culture to the starting $OD_{600}$ of 0.005, and grown for 3 h in LB until reaching $OD_{600}$ of 0.3. The cultures corresponding to each timepoint were grown in individual flasks. To ensure comparable total cell numbers and final amount of protein extracts in treated and untreated cultures, the cells were grown in different volumes. The total volumes of each culture were as follows (timepoint – ml): T0 – 350, T1 – 200, T2 – 500, T3 – 100, T4 – 200. For harvesting cells were transferred on ice into 400 ml centrifuge bottles, washed twice with PBS by centrifugation at $32,600 \times g$ for 20 min at 4°, resuspended in lysis buffer (final concentration 0.8% NP-40, 1.5 mM $MgCl_2$, protease inhibitor, phosphatase inhibitor, 0.4 U/µl benzonase), transferred to 2 ml tubes supplemented with glass beads. The cells were lysed by 2 cycles of 2 min vortexing at 4 °C to avoid sample heating. Cell debris was removed by centrifugation at $15,000 \times g$ for 10 min at 4 °C. The supernatant was diluted to 1 mg/ml protein concentration and distributed by 0.1 mg in PCR tubes. Each aliquot was heated for three minutes to a different temperature (37.0-40.4-44.0-46.9-49.8-52.9-55.5-58.6-62.0-66.3 °C). Protein aggregates were removed by centrifugation at $80,000 \times g$ for 40 min at 4 °C. Protein concentration in the supernatant was determined, and 10 µg protein (based on the two lowest temperatures) was used for further sample preparation. Proteins were reduced, alkylated, and digested with trypsin/Lys-C using the S-trap from OASIS. Peptides were labeled with TMT10plex (Thermo Fisher Scientific). Samples were combined in

the following manner: experimental conditions (incl. control, T1, T2, T3, T4) were combined for two subsequent temperature steps, resulting in five batches to be analyzed per experiment. In addition, both samples from NP-40 lysis (37 °C) and SDS lysis from each experimental condition were combined, allowing the determination of protein solubility differences. The total number of batches per biological replicate (per respective strain) was six, with two replicates each, amounting to 12 measurements. For four biological replicates and two respective strains, K-12 and ED1a, the total number of MS-measured samples was 96.

## LC-MS/MS measurement

Peptides were separated using an UltiMate 3000 RSLC nano LC system (Thermo Fisher Scientific) equipped with a trapping cartridge (Precolumn C18 PepMap 100, 5 µm, 300 µm i.d. x 5 mm, 100 Å) and an analytical column (Acclaim PepMap 100, 75 µm x 50 cm C18, 3 µm, 100 Å). The LC system was directly coupled to an Orbitrap Eclipse mass spectrometer (Thermo Fisher Scientific) using a Nanospray-Flex ion source. Solvent A was 99.9% LC-MS grade water (Thermo Fisher Scientific) with 0.1% formic acid and solvent "B" was 99.9% LC-MS grade acetonitrile (Thermo Fisher Scientific) with 0.1% formic acid. Peptides were loaded onto the trapping cartridge using a flow of 30 µL/min of solvent A for 3 min. Peptide elution was afterward performed with a constant flow of 0.3 µL/min using a total gradient time of 120 min. During the elution step, the percentage of solvent B was increased stepwise: 2% to 4% B in 4 min, from 4% to 8% in 2 min, 8% to 28% in 96 min, and from 28% to 40% in another 10 min. A column cleaning step using 80% B for 3 min was applied before the system was set again to its initial conditions (2% B) for re-equilibration for 10 min.

The peptides were introduced into the mass spectrometer Orbitrap Eclipse (Thermo Fisher Scientific) via a Pico-Tip Emitter 360 µm OD x 20 µm ID; 10 µm tip (New Objective). A spray voltage of 2.3 kV was applied and the mass spectrometer was operated in positive ion mode. The capillary temperature was 320 °C. Full-scan MS spectra with a mass range of 375−1200 $m/z$ were acquired in profile mode in the Orbitrap using a resolution of 70,000. The filling time was a maximum of 250 ms, and/or a maximum of 3e6 ions (automatic gain control, AGC) was collected. The instrument was cycling between MS and MS/MS acquisition in a data-dependent mode and consecutively fragmenting the Top 10 peaks of the MS scan. MS/MS spectra were acquired in profile mode in the Orbitrap with a resolution of 35,000, a maximum fill time of 120 ms and an AGC target of 2e5 ions. The quadrupole isolation window was set to 1.0 $m/z$ and the first mass was fixed to 100 $m/z$. The normalized collision energy was 32 and the minimum AGC trigger was 2e2 ions (intensity threshold 1e3). Dynamic exclusion was applied and set to 30 s. The peptide match algorithm was set to 'preferred' and charge states 'unassigned', 1, 5−8 were excluded.

## MS-data processing and analysis

All proteomics data were processed in Thermo Proteome Discoverer 2.4 using a modified SPS-MS3-TMT template (full parameter sets available on PRIDE, PXD044697). In brief, spectra were matched to an *E.coli* sequence database downloaded from Uniprot (K12 and ED-1a, 08/2020) and to contaminant and decoy databases using the Sequest HT node. Tryptic peptides with 0 to 2 missed cleavages were included, and precursor mass tolerance was set to 10 ppm and fragment ion tolerance to 0.6 Da. Static modifications included TMT-labeling (N-terminus) and Carbamidomethylation (Cys) and dynamic modifications included oxidation of Methionine. Search results were further processed using the Percolator node with $q$-value based validation and concatenated target/decoy selection and FDR targets of 0.01 (strict) and 0.05 (relaxed). Reporter ions were quantified using HCD-MS3 with 20ppm integration tolerance. TPP data were further processed and analyzed according to ref. 19 Statistical tests were applied according as indicated in the respective figure legends (Fisher-Exact test for GO-enrichment, $t$ test for differences in abundance and stability scores between strains).

## Cell cryo plunge freezing and cryo-FIB milling

For vitrification of *E. coli* cells, 3.5 µl cell suspension were taken directly from the cell cultures and pipetted on Quantifoil R 1/1 grids glow discharged for 90 s at 15 mA and 0.38 mbar using PELCO easiGlow. Grids were transferred to an EM GP2 plunge freezer (Leica Microsystems), excess liquid was blotted for 10 s at 20 °C, grids were plunge frozen in liquid ethane and clipped into Autogrid sample holders. Autogrids were mounted into a FIB-shuttle and transferred using a cryo-transfer system to the cryo-stage of a dual-beam Aquilos FIB-SEM (Thermo Scientific). Samples were coated with an organometallic platinum layer using a gas injection system for 10 s and additionally sputter-coated with platinum at 1 kV and 10 mA current for 10 s. SEM imaging was performed at 2−10 kV and 13 pA. Milling was performed by a stepwise current decrease from 500 to 100 pA. Lamellae were polished at 30−50 pA beam currents at shallow angles of 7−9°. Prior to grid unloading some lamellae were sputter-coated with platinum 1−2 s at 1 kV and 10 mA. Grids were stored in liquid nitrogen until data collection.

## Cryo-ET data collection

Cryo-ET data were collected on a Titan Krios G2 transmission electron microscope (Thermo Scientific) operated at 300 kV and equipped with a BioQuantum-K3 energy filter (Gatan). The K3 direct electron detector (Gatan) was operated in electron counting mode. Montages of individual lamellae were acquired at 6500 X magnification with a 2.83 nm pixel size. Tilt series (TS) projections were acquired at a magnification of 53,000X, corresponding to a pixel size 1.697 Å at the specimen level over a tilt range of −63° to 45°, starting from an pre-tilt angle of 7−9° to compensate for lamella pre-tilt relative to the grid, at 3° tilt increment, following a dose-symmetric scheme[50], with tilt increments grouped by 2. The defocus range was set to −1.5 to −5 µm. Total dose per tilt series was set to ~ 120 e⁻/Å², a 70 µm objective aperture was inserted and energy slit width was set to 20 eV for TS collection. Tilt series images were acquired as 6 K x 4 K movies of 10 frames each, which were motion-corrected on-the-fly in SerialEM[51]. In total, 87 TS were acquired for *E. coli* K-12 LB, 82 TS for *E. coli* K-12 TET, 51 TS for *E. coli* ED1a LB, and 44 TS for *E. coli* ED1a TET samples. All of the cryo-ET data were collected for the timepoint of 30 min.

## Tilt series alignment and template matching

Motion-corrected tilt series were visually inspected to remove bad projections at high tilts. Then the TS were aligned through patch-tracking in IMOD[52] and reconstructed using weighted back-projection with SIRT-like filtering of 10 iterations at a pixel size of 13.576 Å. These tomograms were used for visual inspection of tilt series alignment and tomogram thickness. Selected TS and their respective tomograms were reconstructed in Warp[53] based on the IMOD-derived alignment files. Cross-correlation-based template matching in the Dynamo package[54] was used to localize ribosome particles within reconstructed tomograms. To prepare the ab initio-like reference, five tomograms generated in IMOD with SIRT-like filtering were matched using angular sampling of 30° and 70S ribosome (EMDB-4050) as a template. The template was downsampled to 13.576 Å per voxel and filtered to ~ 70 Å. The highest 1000 cross-correlation peaks were extracted as sub-tomograms at 6.788 Å pixel size. To avoid duplicates, after extraction of every particle, a 300 Å mask in diameter was applied to the cross-correlation volume. Subtomograms were aligned using NovaSTA package (Turoňová, Zenodo, 2022 https://github.com/turonova/novaSTA). The obtained structure was used as a reference for template matching of all the datasets in the Warp-generated tomograms. For consequent STA and classification, particles were extracted as described above. The particles list was converted to a Warp-compatible

star-file using the *dynamo2m* toolbox (https://github.com/alisterburt/dynamo2m)[55]. The summary of data collection parameters and image processing statistics is given in Supplementary Table 2.

### Intermembrane distance measurement

Intermembrane distances were measured in the cells that were used for cryo-ET and subtomogram averaging. The calculations were performed on the visible part of the cell envelope extracted directly from the tilt series collected at 53,000 X magnification. For each tilt series, one image near lamella pre-tilt angle was selected. Membranes were segmented in Fiji[56] followed by application of a 3-nm Gaussian filter and 10-nm variance filter. The distances between two membrane leaflets were measured at points spaced by 10 nm using a script (https://github.com/martinschorb/membranedist) and manually inspected to remove obvious outliers (Supplementary Fig. 4e).

### Subtomogram averaging and classification

The datasets corresponding to each of the four conditions, namely K-12/ED1a 30 min LB/TET were initially processed separately. The step-by-step processing scheme is described in Supplementary Fig. 6. Subtomograms were extracted with a pixel size of 6.788 Å and a box size of 64 pixels. The initial average was generated from the extracted subtomograms using the *relion_reconstruct* command and was used as a reference for the rough 3D classification in Relion 3.1[57]. The ribosomes were sorted from the membranes and junk particles using 3D classification with the following parameters: 7.5° angular sampling, offset range 5 pixels, offset search step 1 pixel. To resolve the high-resolution subtomogram average of the *E. coli* 70S ribosome, the 70S particles from all datasets were merged and aligned in Relion and then refined using M[44] at a pixel size of 6.788 Å. The particles were extracted with 3.394 Å/px and a box size of 128 pixels, refined in Relion, and subjected to the focused 3D classification. The 75,670 well-aligned particles were re-extracted with 1.697 Å/px and a box size of 294 pixels and refined using 3 iterations in M to a final resolution of 5.3 Å.

To sample ribosomes states, 70S subtomograms were classified focusing on P-site tRNA and factor-binding regions without alignment. The resulting classes representing ribosomes at different states were refined in M. The following parameters were used for all 3D classification jobs: reference low-pass filter 50 Å, number of classes 3-6; T-parameter 10; 25−35 iterations; 350 Å mask diameter After every classification step, the selected classes were refined in Relion. The star files from Relion were converted to M-compatible format using the *relion_downgrade* command of the *dynamo2m* package.

### Reporting summary

Further information on research design is available in the Nature Portfolio Reporting Summary linked to this article.

### Data availability

Cryo-ET density maps generated in this study have been deposited in the EM Data Bank with the following accession codes: EMD-18036 (in situ *E. coli* 70 S ribosome), EMD-18037 (in situ 70 S ribosome of *E. coli* K-12 untreated cells), EMD-18038 (in situ 70 S ribosome of *E. coli* K-12 cells treated with tetracycline), EMD-18039 (in situ 70 S ribosome of *E. coli* ED1a untreated cells), EMD-18040 (in situ 70 S ribosome of *E. coli* ED1a cells treated with tetracycline), EMD-18041 (*E. coli* K-12 70 S ribosome bound to mRNA A-tRNA, P-tRNA, E-tRNA), EMD-18042 (*E. coli* ED1a 70 S ribosome bound to mRNA A-tRNA, P-tRNA, E-tRNA), EMDB-19206 (*E. coli* ED1a 70 S·TET 30 S head focused), EMDB-19207 (*E. coli* ED1a 70 S·TET 30 S body focused), EMDB-19208 (*E. coli* ED1a 70 S·TET 50 S focused). Cryo-ET raw tilt series have been deposited in the EM Public Image Archive (EMPIAR) with the following accession codes: EMPIAR-11945 (*E. coli* K-12 untreated cells), EMPIAR-11946 (*E. coli* K-12 tetracycline-treated cells), EMPIAR-11947 (*E. coli* ED1a untreated cells),

EMPIAR-11948 (*E. coli* ED1a tetracycline-treated cells). The ribosome models that were used for interpretation of obtained maps and structural comparisons were downloaded from the protein data bank (PDB) under following accession numbers: 4V9A, 5J5B, 7K00, 8CF1. All proteomics data associated with this manuscript have been uploaded to the PRIDE online repository[58] (https://www.ebi.ac.uk/pride/) with the identifier PXD044697. Source data are provided with this paper.

### Code availability

Script for measuring intermembrane distances is available at the repository https://github.com/martinschorb/membranedist. Scripts used for mass spectrometry data analysis can be found at https://doi.org/10.5281/zenodo.10979011.

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

## Acknowledgements

The authors would like to acknowledge Özkan Yildiz, Juan F. Castillo Hernandez, Thomas Hoffmann for support with scientific computing, Mark Linder and the Central Electron Microscopy facility of the Max Planck Institute of Biophysics for support with cryo-ET sample preparation and data acquisition. We thank Barbara Rathmann for assistance in MS data collection. The authors are grateful to Stefanie Böhm for critical reading and assessment of the manuscript. I.K. acknowledges Wim Hagen, and Felix Weiss for initial electron microscopy training; Matteo Allegretti, Maximilian Seidel, Patrick Hoffmann, and Andre Schwarz for discussions on the project. M.B. acknowledges funding by the Max Planck Society and EMBL.

## Author contributions

I.K. conceived the project, designed experiments, performed experiments, analyzed data, and wrote the manuscript. N.R. developed the TPP analysis pipeline, analyzed data, and wrote the manuscript. C.G. and A.T.

conceived the project, advised and provided protocols for microbiology experiments, edited the manuscript. B.T. and C.E.Z. supported cryo-ET data processing. S.W assisted cryo-ET data acquisition. J.D.L. supervised mass spectrometry experiments, M.B. conceived the project, supervised the project, analyzed data, acquired funding, and wrote the manuscript.

## Funding

## Competing interests
The authors declare no competing interests.
