## [Peer Review File · Nature Communications]

Bactericidal effect of tetracycline in *E. coli* strain ED1a may be associated with ribosome dysfunctionEditorial Note: This manuscript has been previously reviewed at another journal that is not operating a transparent peer review scheme. This document only contains reviewer comments and rebuttal letters for versions considered at *Nature Communications*.

REVIEWER COMMENTS

Reviewer #1 (Remarks to the Author):

First of all I wish to clarify that I did not criticize the initial version of this manuscript for not yielding the elucidation of the TET mechanism of action. My impression was instead: (1) that it had some overclaims or shortcut conclusions that were not supported by the data and (2) that overall it was unclear how this new data integrates and what this brings toward the future elucidation of TET mechanism.

The new version of the manuscript is more tailored to the data. The authors proposed that their data exclude some possible mechanisms of action and bring observations on the different consequences of TET treatment on the 2 bacterial strains studied. However the data is collected with a 2-fold difference in intracellular antibiotic concentration in the 2 strains to be compared (details below). In my opinion, this makes it difficult to reliably interpret the data of this work and there is a risk that the current data reflects this difference in internal TET concentration, rather than new insights towards understanding the differential TET mechanism of action in the 2 strains. In addition, with a single antibiotic and a single timepoint (which is a rather late one for ED1 in particular), it is difficult to discriminate which observations are possible part of TET mechanism of action and which are a consequence of cells starting to die.

Main points

The intracellular TET concentrations in both strains differ by a factor of 2 (according to the authors data Fig 2D). The authors answered that: “the amount of TET is orders of magnitude in excess over the ribosome, even at the lower concentration”.

Do the authors have evidence that the actual absolute intracellular TET concentration indeed corresponds to large excess of TET compared to ribosomes?

The intracellular concentration of an antibiotic is regulated by many mechanisms and I don't think that the assumption of TET large excess can be made without actual data to measure these absolute intracellular concentrations.

In the absence of this piece of data and to have a rough idea of orders of magnitudes, one can attempt a calculation with intracellular concentration = concentration applied to the medium: 8 ug/mL. This results in about 10 000 TET molecules that would be present in one E coli cell (eg. in ED1a, and about half so 5 000 TET molecules for K12 strain). Each E coli cell has at most ~14 000 ribosomes in mid log phase.

Hence, it is not obvious that TET would be in order of magnitude excess when compared to ribosome numbers and a difference of a factor of 2 between the 2 cell lines could actually have a non-negligible impact in terms of ribosome binding and downstream effect. At this concentration range for instance about half of the ribosomes of K12 may be bound by TET while nearly all ribosomes of ED1a would be. This could definitely largely contribute to a higher thermostabilization of ED1a ribosomes observed by TPP and is also likely to precipitate cell death in ED1a which would consequently result in membrane destabilization and shut down of translation as observed by cryoET.

I understand the point that TET remains bacteriostatic in K12 even when higher concentration of drugs are internalised, hence the difference in internal drug concentration may not be the only reason explaining the different effect of the antibiotic on cell viability in general. However to address the question 'how does TET remain bacteriostatic at high internal concentration in K12?', one would presumably need to look at what happens in K12 cells at these high concentrations, rather than extrapolating from one datapoint at a lower concentration. In addition, the fact that TET remains bacteriostatic at high concentration in K12 does not mean that applying a 2-fold difference in lower intracellular concentrations to the 2 strains will not result in different consequences on intracellular molecular measurements as performed here.

All experiments in this study (TPP, cryoET etc.) are done with this 2 fold difference on intracellular Tet concentration. So even if Tet was in excess (where K_d of the binding reaction etc. would also need to be taken in consideration), without the control data at similar intracellular concentrations, I don't see how it is possible to exclude that this 2-fold difference in Tet intracellular concentration is responsible for some of the observed differences in the various experiments.

Another limitation of this work is that, with a single antibiotic and a single timepoint (that is rather late for ED1 in particular), it is difficult to conclude that the differences observed, for instance on membrane destabilization and on ribosome actively elongating or not, are a cause of the different action of TET rather than a consequence of the ED1 cells starting to die.

Other point

The authors added the single particle analysis visualization of TET binding to the ED1a ribosome, confirming that it is identical to the binding previously described for K12. This was obtained by mixing idle (no tRNA) purified ribosomes of untreated cell with TET. However the authors explain that in the

cells most ribosomes are typically bound by tRNA so in order to look at the TET binding site in a more relevant ribosome conformation they use a crude purification of ribosomes from untreated K12 and ED1a and they focus their analysis on a state containing A,P and E tRNAs. However this state is not found in the in situ data, which they explain by the fact that it is likely transient in cells. But then why choose this state for analysis in the SPA data, knowing that this is not one of the abundant relevant ones in the cell - and hence it is quite unlikely that a TET molecule would encounter a ribosome in this state in a cell at a given time? Is it because the 70S-Ef-Tu-A/T,P,E state (observed in cryoET) was absent from SPA data? It is also still unclear to me what this analysis done on untreated cells may bring to our understanding of TET binding to these ribosomes.

Reviewer #2 (Remarks to the Author):

After carefully reading the revised version of the manuscript as well as the responses of the authors to the reviewer's comments, I can say that the authors improved the manuscript and addressed some of the critical points raised by this and other reviewers.

Response:

We have revised the text to clarify any statements that might have been confusing. Modifications are highlighted yellow in the main text.

REVIEWER COMMENTS

Reviewer #1 (Remarks to the Author):

First of all I wish to clarify that I did not criticize the initial version of this manuscript for not yielding the elucidation of the TET mechanism of action. My impression was instead: (1) that it had some overclaims or shortcut conclusions that were not supported by the data and (2) that overall it was unclear how this new data integrates and what this brings toward the future elucidation of TET mechanism.

The new version of the manuscript is more tailored to the data. The authors proposed that their data exclude some possible mechanisms of action and bring observations on the different consequences of TET treatment on the 2 bacterial strains studied. However the data is collected with a 2-fold difference in intracellular antibiotic concentration in the 2 strains to be compared (details below). In my opinion, this makes it difficult to reliably interpret the data of this work and there is a risk that the current data reflects this difference in internal TET concentration, rather than new insights towards understanding the differential TET mechanism of action in the 2 strains. In addition, with a single antibiotic and a single timepoint (which is a rather late one for ED1 in particular), it is difficult to discriminate which observations are possible part of TET mechanism of action and which are a consequence of cells starting to die.

Main points

The intracellular TET concentrations in both strains differ by a factor of 2 (according to the authors data Fig 2D). The authors answered that: "the amount of TET is orders of magnitude in excess over the ribosome, even at the lower concentration".

Do the authors have evidence that the actual absolute intracellular TET concentration indeed corresponds to large excess of TET compared to ribosomes?

The intracellular concentration of an antibiotic is regulated by many mechanisms and I don't think that the assumption of TET large excess can be made without actual data to measure these absolute intracellular concentrations.

Response:

Since the reviewer and the authors agree that the exact intracellular concentration is determined by a complex mechanism, we have to admit that we cannot provide such evidence. As previously pointed out, we do not think that this is relevant, for two reasons. 1. Our experimental design is suitable for the question asked how do both strains react to the exact same antibiotic stress. An experiment in which the intracellular Tet concentration is the same and stable over time, is not straight forward to design. 2. We have not claimed that the phenotype is caused by a differential mechanism of drug binding to the ribosome. Our conclusions explicitly do not exclude the possibility that an unknown mechanism has led to the differential intracellular Tet fluorescence measured. We thus do not think that we overclaim anything in the revised version of the manuscript.

However, our analysis nonetheless shows that the two strains respond differently on a molecular level and that in situ analyses are crucial here, as only assessing the binding of TET to the respective purified ribosomes would not have yielded a difference.

In the absence of this piece of data and to have a rough idea of orders of magnitudes, one can attempt a calculation with intracellular concentration = concentration applied to the medium: 8 ug/mL. This results in about 10 000 TET molecules that would be present in one E coli cell (eg. In ED1a, and about half so 5 000 TET molecules for K12 strain). Each E coli cell has at most ~14 000 ribosomes in mid log phase.

Hence, it is not obvious that TET would be in order of magnitude excess when compared to ribosome numbers and a difference of a factor of 2 between the 2 cell lines could actually have a non-negligible impact in terms of ribosome binding and downstream effect. At this concentration range for instance about half of the ribosomes of K12 may be bound by TET while nearly all ribosomes of ED1a would be. This could definitely largely contribute to a higher thermostabilization of ED1a ribosomes observed by TPP and is also likely to precipitate cell death in ED1a which would consequently result in membrane destabilization and shut down of translation as observed by cryoET.

Response:

We agree that it is impossible to know the exact intracellular TET concentration, as pointed out above however, this does not affect the conclusions of our manuscript.

I understand the point that TET remains bacteriostatic in K12 even when higher concentration of drugs are

internalised, hence the difference in internal drug concentration may not be the only reason explaining the different effect of the antibiotic on cell viability in general. However to address the question 'how does TET remain bacteriostatic at high internal concentration in K12?', one would presumably need to look at what happens in K12 cells at these high concentrations, rather than extrapolating from one datapoint at a lower concentration. In addition, the fact that TET remains bacteriostatic at high concentration in K12 does not mean that applying a 2-fold difference in lower intracellular concentrations to the 2 strains will not result in different consequences on intracellular molecular measurements as performed here.

Response:

Yes, we agree that this could occur by a different mechanism. However, the question posed by the reviewer 'how does TET remain bacteriostatic at high internal concentration in K12?', is not the focus of our manuscript.

All experiments in this study (TPP, cryoET etc.) are done with this 2 fold difference on intracellular Tet concentration. So even if Tet was in excess (where K_d of the binding reaction etc. would also need to be taken in consideration), without the control data at similar intracellular concentrations, I don't see how it is possible to exclude that this 2-fold difference in Tet intracellular concentration is responsible for some of the observed differences in the various experiments.

Response:

As pointed out above, we do not want to exclude that the observed '2 fold difference on intracellular Tet concentration' has an effect. We had not made any respective claims in the revised version.

Another limitation of this work is that, with a single antibiotic and a single timepoint (that is rather late for ED1 in particular), it is difficult to conclude that the differences observed, for instance on membrane destabilization and on ribosome actively elongating or not, are a cause of the different action of TET rather than a consequence of the ED1 cells starting to die.

Response:

As pointed out in the previous response, we do not claim that we have fully determined the molecular effect of TET on ED1a (or K12) and we do not disagree with the reviewer that some of the phenotypes, such as the membrane destabilization, we observe after 30min treatment may be indirect consequences of TET's primary effect on cells. However, these phenotypes are ultimately induced by TET treatment and as the ribosome is its major target in cells, we find it plausible that the altered distribution of ribosomal states would likely be attributed to TET binding.

Other point

The authors added the single particle analysis visualization of TET binding to the ED1a ribosome, confirming that it is identical to the binding previously described for K12. This was obtained by mixing idle (no tRNA) purified ribosomes of untreated cell with TET. However the authors explain that in the cells most ribosomes are typically bound by tRNA so in order to look at the TET binding site in a more relevant ribosome conformation they use a crude purification of ribosomes from untreated K12 and ED1a and they focus their analysis on a state containing A,P and E tRNAs. However this state is not found in the in situ data, which they explain by the fact that it is likely transient in cells. But then why choose this state for analysis in the SPA data, knowing that this is not one of the abundant relevant ones in the cell - and hence it is quite unlikely that a TET molecule would encounter a ribosome in this state in a cell at a given time?

Is it because the 70S-Ef-Tu-A/T,P,E state (observed in cryoET) was absent from SPA data?

It is also still unclear to me what this analysis done on untreated cells may bring to our understanding of TET binding to these ribosomes.

Response:

In the initial version of our paper, we had not included a structure of ED1a ribosomes bound to TET, because the genetic information indicated that all the rRNA nucleotides engaged in the TET binding are actually the same in both strains, and the K12 ribosome structure was available (Cocozaki et al., 2016). This is mentioned in the manuscript. Due to the reviewer's comments in the first round of revision, we also solved the ribosome structure from the ED1a strain bound to TET. The result was exactly as we had anticipated, all rRNAs in the structure that could engage with TET, are identical in structure. We thus conclude that the question was addressed, namely "confirming that it is identical to the binding previously described for K12".

As many studies have shown meanwhile, a consistency between the functional states observed in vitro and in situ cannot be expected, because isolation stalls ribosomes in particular states (Behrmann et al., 2015; Fromm et al., 2023; Hoffmann et al., 2022; Xing et al., 2023; Xue et al., 2022). Thus, for SPA of crude ribosome extract we chose a condition in which A-site tRNA engagement "conveys structural stability to the TET-binding region" and opted for the experimental strategy mentioned by the reviewer.

Reviewer #2 (Remarks to the Author):

After carefully reading the revised version of the manuscript as well as the responses of the authors to the reviewer's comments, I can say that the authors improved the manuscript and addressed some of the critical points raised by this and other reviewers.

Response:

We thank the reviewer for again reviewing our manuscript and the positive comments.

References:

- Behrmann, E., Loerke, J., Budkevich, T.V., Yamamoto, K., Schmidt, A., Penczek, P.A., Vos, M.R., Burger, J., Mielke, T., Scheerer, P., *et al.* (2015). Structural snapshots of actively translating human ribosomes. *Cell* **161**, 845-857.
- Cocozaki, A.I., Altman, R.B., Huang, J., Buurman, E.T., Kazmirski, S.L., Doig, P., Prince, D.B., Blanchard, S.C., Cate, J.H., and Ferguson, A.D. (2016). Resistance mutations generate divergent antibiotic susceptibility profiles against translation inhibitors. *Proc Natl Acad Sci U S A* **113**, 8188-8193.
- Fromm, S.A., O'Connor, K.M., Purdy, M., Bhatt, P.R., Loughran, G., Atkins, J.F., Jomaa, A., and Mattei, S. (2023). The translating bacterial ribosome at 1.55 Å resolution generated by cryo-EM imaging services. *Nat Commun* **14**, 1095.
- Hoffmann, P.C., Kreysing, J.P., Khusainov, I., Tuijtel, M.W., Welsch, S., and Beck, M. (2022). Structures of the eukaryotic ribosome and its translational states in situ. *Nat Commun* **13**, 7435.
- ribosomes. *Nat Struct Mol Biol* **30**, 1380-1392.
- Xing, H., Taniguchi, R., Khusainov, I., Kreysing, J.P., Welsch, S., Turoňová, B., and Beck, M. (2023). Translation dynamics in human cells visualized at high-resolution reveal cancer drug action. 2023.2003.2002.529652.
- Xue, L., Lenz, S., Zimmermann-Kogadeeva, M., Tegunov, D., Cramer, P., Bork, P., Rappsilber, J., and Mahamid, J. (2022). Visualizing translation dynamics at atomic detail inside a bacterial cell. *Nature* **610**, 205-211.